



# Wind Influences Corrected Auto-calibrated Soil Evapo-Respiration Chamber (ASERC), Evaporation Measures.

Bartosz M. Zawilski

CESBIO Université de Toulouse, CNES, CNRS, INRA, IRD, UPS, Toulouse, 31000 France

*Correspondence to*: Bartosz M. Zawilski (zawilskib@cesbio.cnes.fr)

**Abstract.** The importance of the soil evaporation concerns our main life supports source for agriculture or for climate changes predictions science. A simple to operate instrument, based on non-steady state (NSS) technique, made for soil evaporation measurement appears then suitable. However, because the NSS chamber technique is highly invasive, special care should be

provided to correct the wind influence on the evaporation process. As the wind influence on the evaporation is depending on numerous and not real-time monitorable variables, in order to make the measurements easily corrigible on a bare soil with a unique variable - wind speed (Ws), whatever is the soil nature, soil texture, and others soil or air meteorological variables - a self-calibrating chamber with corresponding protocol called Auto-calibrated Soil Evapo-Respiration Chamber (ASERC) was developed. A simple protocol followed by this chamber allows to determine the soil evaporation wind susceptibility (Z) and

to correct the measurements achieving 0.95 accuracy confidence. Some interesting finding on sandy and clayey soils evaporation measured during a laboratory calibration will also be reported.

**Introduction**. In the context of global temperature rising, as the water is our main life support key resource to the food production and the water vapor is one of the most abundant greenhouse gases in the Earth atmosphere, it is important to gather

knowledge about soil evaporation. Soil evaporation may be a major soil moisture loss source. On the one hand, the global direct annual soil evaporative precipitations lose are as high as 20% and the other 40% though the vegetation transpiration (Oki and Kanae 2006), and the soil evaporation may reach up to 75% in the arid and semi-arid regions (WMO168 2008) when the total soil evaporation along with the vegetation transpiration, so-called evapotranspiration, may dissipate up to 90% of the annual rain fall (Pilgrim et all 1988, Wilcox et all. 2003). On the other hand, the soil evaporation consumes about 20% of the

solar radiation energy (Trenberth et al. 2006). Energy absorbed on the soil surface or in the soil subsurface during the evaporation process, lowering down soil temperature, released later in the higher atmosphere layer when condensing, warming up the air. The water vapor is the most important greenhouse gas in the atmosphere not because of its efficiency but because it is the most present; 60% of the total greenhouse effect (Trenberth et al. 2009, Schmidt et all. 2010) and its recently measured upper-tropospheric concentration increase is directly attributable to the human's activities (Choung et al. 2014). Good et al.

(2015) shows that the main water vapor source (65%) is the soil and not surface waters. Motivated especially by measured soil energy budget imbalance of a notable importance and a probable subsurface evaporation contribution, a simple and versatile



soil evaporation measurement instrument was developed based on a Non-Steady State (NSS) automatic chamber technique, with special attention given to the solar radiation heating screening and the wind influence corrections. Chamber construction and it characteristic such as real mixing time assessment, developed protocol for evaporation calculations, and wind influence

corrections will be reported. This study is based on over 1000 measurement cycles (that is over 10 000 chamber deployments during over two years) and, after calibration of wind influence on the evaporation and chamber perturbations correction, showing a reasonable agreement between chamber measurements and real evaporation with $R^2 > 0.95$. The same correction formula is used for sandy and clayey soils whatever is the soil moisture.

## 1 Existing evaporation measurement techniques

Widely used eddy covariance technique is a relatively expensive but little invasive way to estimate the soil evaporation on the bare soil and other trace gases fluxes. As every measurement technique, eddy covariance has its pro and cons. Eddy covariance provides the evaporation estimation when the air flow is turbulent enough that means only when the wind is strong enough which is usually not the case at night. Also, these measurements are not precisely localized and the provenance is approximative. This point is often a force since the measurements reflecting the mean process but not when a precise

provenance is sought-after. However, eddy covariance cannot be implemented everywhere. The site should be flat and big enough, please see edited by Aubinet et al. (2012) an eddy covariance dedicated book which is describing this technique and its requirements from practical point of view. Also, a systemic underestimation of eddy covariance CO2 flux comparatively to closed chambers techniques given fluxes was pointed out by numerus authors (Goulden et al. 1996, Norman et al. 1997, Law et al. 1999, Hollinger et al. 1999, Janssens et al. 2000, Pavelka et al. 2007, Zha et al. 2007, Myklebust et al. 2008, Schrier-Uijl

et al. 2010). Nevertheless, we have to kept in mind that chambers measurements have to be carefully considered and, in the same way as for evaporation, other gases efflux measurements have to be corrected versus the wind (Zawilski in progress). In the calm conditions, due to the internal fan perturbations, closed chambers may have tendency to overestimate a possible soil efflux (Schneider et al. 2009, Brændholt et al. 2017).

In the case of vegetation presence, in order to separate transpiration from soil evaporation, an experimental isotopic mass balance approach can be adopted (Ferretti et al. 2003) which requires a frequent soil sampling and laboratory analyze or expensive and voluminous analyzer use. Eddy covariance measurements giving a total (soil evaporation and vegetation transpiration) evapotranspiration coupled with a partitioning accordingly to a model (see Koola et al. 2014 for review) are performed to separate soil and vegetation contribution. Each specific model may be accurate but only for a specific plant,

making difficult to apply it to a mixed plant cover. Moreover, even for a specific plant, there are numerous models giving different results. Nothing that for the maize, there are over 29 models (Kimball et al. 2019). Also, each model requires more or less numerous variable injections which make some models difficult to be applied since the required variables are not known (Kustas and Agam 2014).



Another widely used technique for direct soil evaporation measurements is the lysimeters from a micro-lysimeters for bare

soil evaporation measurement to a large scale lysimeters in order to measure the total evapotranspiration (listed by Liu et all 2002.). This technique consists roughly in weighting of a hold soil colon giving a direct evaporation or evapotranspiration measurement if the surface of the soil colon is large enough to hold supplementary vegetation. However, this apparatus should be buried deeply making relatively hard to implement this measurement especially when frequent apparatus displacements are necessary as it is the case on an agricultural plot (tillage and others soils operations). Lysimeters require also a deep enough

soil which is not the case with the shallow rock presence and even simple stones presence, and provide a timely averaged measurement because the weigh variation caused by water evaporation needs to be important enough comparatively to the total enclosed soil weigh.

Relatively recently, a heat balance-based method using the heat pules probes was proposed by Sauer et al. (2007) and Heitman et al. (2008 and 2017). This technique is based on assumption that the heat budget disclosure is only due to the water

evaporation and allows measurements only in the subsurface leaving the surface evaporation unmeasurable. Yet, this technique is the only one which allows tracking the depths of the subsurface evaporation.

Therefore, an exclusive, fast and easy to implement total bare soil evaporation measurement is suitable. A dynamic closed Non-Steady State (NSS) technique, used for soil effluxes measurements may be also used for soil evaporation assessment. However, for these measurements as for other soil effluxes measurements, measured data have to be carefully corrected.

Indeed, this technique is not a direct measurement and is highly invasive. Not only the collar inserted into the soil may perturb the comportment of the enclosed soil by root shearing (Heinemeyer et al. 2011) limiting the autotrophic component of the respiration, but also just by the chamber presence and more particularly during the chamber head deployment with the enclosed part of the soil isolation from the exterior meteorological conditions such as the wind (see Rochette et al. 1997 and Rochette and Hutchinson 2005 for chamber technique description).






## 2 Materials and methods

### 2.1 Chamber construction

The chamber described later in this paper was constructed in the laboratory (please see Fig. 01).

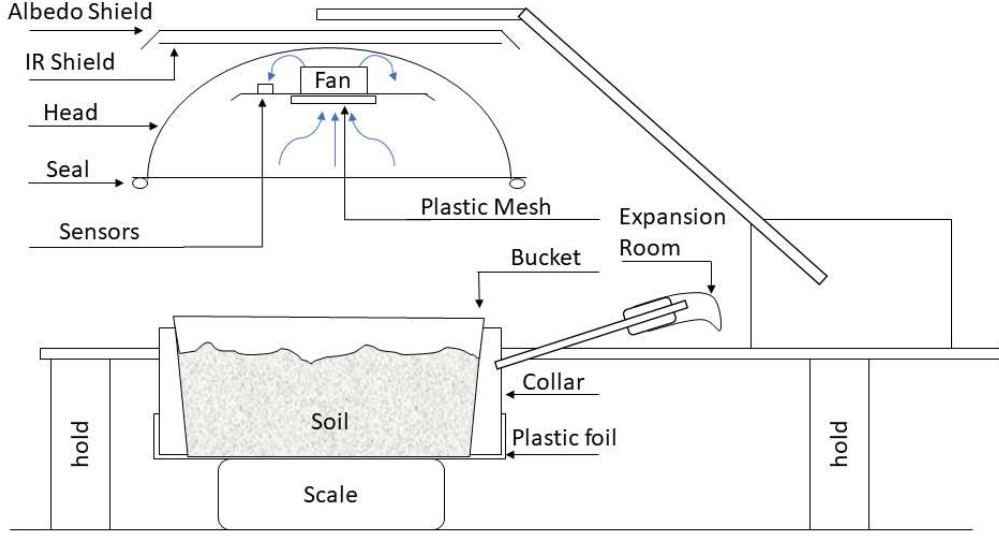


**Figure 01: Scheme of the chamber and experimental setup. In this sketch the chamber head is not deployed.**

When the chamber head is deployed, the cloche with embedded fan is firmly put down on the base insulating the collar with the bucket and a well delimited air portion. Inside this finite air volume, due to the soil evaporation, the air moisture *RH* will rise more or less quickly up to reach 100%.

Used internal fan, which is the core of the device, is a Maglev fan PSD1204 PKB 3-A 40x40x20mm (Sunonwealth Electric Machine Industry Company Limited, Qianzhen, District Kaohsiung, Taiwan) with Pulse Wide Modulation (PWM) control and rotation sensor driven by a generic PWM generator able to generate a signal of a given frequency and a given duty on demand and communicating with a datalogger (CR1000 from Campbell Scientific, Logan, Utah, USA.) by a UART (TTL) bus. The main humidity sensor (because there were three of humidity sensors of a different response time for comparison) is a P14

Rapid mounted on a Linpicco plate with a PT1000 sensor for simultaneous humidity and temperature measurements (Innovative Sensor Technology IST AG, Ebnat-Kappel, Switzerland). A pressure, temperature and also humidity again inside the chamber head were monitored using a BME280 (Bosch Sensortec GmbH, Reutlingen, GERMANY) a digital sensor under an Arduino-Uno control (I²C bus) forwarding these measures to the main data logger CR1000 via UART TTL bus. The fan, mounted on a holding plate along with all the sensors, was aspirating the air from the bottom of the chamber head through a



plastic mesh (opening percentage 47%). Not used for this paper but for respiration measurements, an embedded carbon dioxide sensor MH-Z16 (Winsen Electronics Technology Co. Ltd, Zhengzhou, Henan, China) based on a Non-Dispersive Infra-Red (NDIR) technique with a digital interface (UART TTL) under an Arduino-Uno control.

All internal metallic parts were coated with a high PTFE (Teflon) content paint in order to reduce as much as possible the
water vapor sorption on a metallic surface.

Inside the collar was inserted a small pipe allowing the air entrapped by the chamber head, during the chamber deployment, flow freely through to inside a nitrile finger cot (expansion room) in order to equilibrate the inside and outside pressure and allowing a small chamber volume expansion to compensate the mass flow from the soil.


As the heating from solar radiations may strongly affect the evaporation process by artificially rising the chamber head temperature, a special attention was given to shield it with a first albedo shield made from a white painted stainless-steel plate. This shield is screening direct radiations however its temperature will rise and its infra-red radiations may reach the chamber head as well. A well-known technique used for a cryogenics fluids operation was then used by interposing a second Infra-Red
(IR) shield made from a plastic plate coated with an aluminium thick foil on both sides.

All calculations were automatized using a LabView 2015 programming (National Instruments Corporation, Austin, Texas USA)

**2.2 Measurements protocol**

For all the calibration measurements, the chamber was placed on holds and the collar bottom, normally inserted into the soil, was hatched with an elastic plastic foil. An electronic scale was placed just below this foil and a bucket with a studied soil was placed inside the collar reposing on the scale (digital scale WA30002Y from W&J Instrument CO., LTD. Mudu Jiangsu, China) with a continue RS-232 bus output configurated to never turn off (power save disabled) with the plastic foil between
the bucket and the scale plate. A basic scheme depicts chamber and experimental setup function (Fig. 01). With this setup, the bucket mass diminution (enclosed soil water evaporation) was relatively well monitored and provided the real evaporation (*ER*) in the isothermal conditions.

An external fan blows the air on the chamber and an ultrasonic anemometer WindSonic 2D (Gill Instruments Limited,
Lymington Hampshire, UK) allows monitoring and recording on the data logger CR1000 the resulting wind speed 5cm above the sample soil surface. It should be noted that $2ms^{-1}$ measured 5cm above the surface equals to about $6.5ms^{-1}$ measured 2m above the surface (logarithmic profile).

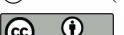



The data logger CR1000 was programmed to command closure (chamber head deployment) or opening of the chamber, to
record the measured humidity and temperature along with the pressure, and the embedded fan rotation speed inside the chamber
head as well as the wind speed before and during the deployment. Before each chamber deployment, a prior 120s flushing with
a 100% duty operated embedded fan was performed. Between each PWM change for measurements, the chamber was opened,
flushed during one minute and then closed again. Every six hours, a measurement cycle was initiated. Any measurement cycle
consists to measure the absolute humidity accumulation in a closed chamber head with the embedded fan powered from
PWM=10% duty to PWM=100% duty by step of 10% giving then ten consecutives chamber deployments. Each chamber
deployment for each PWM takes about 10 minutes. With a flushing time between the deployments, the whole cycle takes over
two hours. The six-hour delay between each measurement cycle was adopted in order to do not strongly perturb the natural
evaporation process. This protocol and ten different PWM measurements are for the fan influence characterization study
purpose only. The real measurement protocol, as described below, is much shorter allowing more frequent chamber
deployment.

The studied soil is either sandy, means a rough sand (0.1-3mm) or clayey (high clay content soil 50% clay, 40% silt, 10%
sand). Real evaporation is deduced by weighting bucked with the studied soil calculating its mass $M_B$ variation provided by
the scale. The mean soil moisture $w$ was determined by the ratio


$$w = \frac{(M_B - M_0)}{M_s}$$

(1)

With $M_0$ being the bucket with dry soil mas and $M_s$ being the dry soil mass (without the bucket mass). This definition,
gravimetric water content, is not a usual volumetric water content but often used for clayey soils since the soil volume changes
and the cracking formation yields to a very complicated volume calculation.

Indeed, an additional difficulty concerns the volume determination for clayey soils as the volume is subject to change with the
soil moisture content (swelling soils). Additionally, the crack apparition makes the volume determination and even its
definition particularly difficult. Are the crack part of the soil sample or not? For these studies, the gravimetric water content is
more usable.

### 2.3 Accumulation calculations

In order to estimate the water vapor efflux by the *NSS* technique, the absolute water vapor concentration $q$ is monitored.
Measured concentration versus time gives a curve regressed by an Exponential Rise (*ER*) formula very widely presents in all



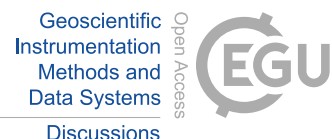

physical process as it is reflecting a variation versus time $dC/dt$, $C$ being the scalar of interest, proportionally to the gap between

the instant scalar value and the stable limit value $C_l$, that the scalar tent to reach:

$$\frac{dC}{dt} = k(C_l - C)$$

(2)

This differential equation has a general solution of the form:


$$C(t) = Ae^{-kt} + B$$

(3)

With $A$ and $B$ being constants.

With initial and final conditions, we can determine these constants using $C_0$ the initial value and $\tau$ the characteristic time also

called response time $(\tau = \frac{1}{k})$ :

$$C(t) = C_l - (C_l - C_0)e^{-t/\tau}$$

(4)

The Evaporation Rate or Measured Evaporation $ME$, by definition, is determined by temporal derivation of the water vapor

concentration $q$:

$$ME = (\frac{dq}{dt})_{t=0}$$

(5)

It is important to note that the water vapor (and others gas emanating from the soil) accumulation inside the deployed chamber

head is not constant. Only the initial evaporation rate (t=0 it means at the chamber head deployment beginning) is retained. In

brief, the scale is allowing the Real Evaporation ($RE$) measurement and sensor monitoring closed chamber head space air

moisture is allowing the Measured Evaporation ($ME$) determination by $ER$ fit and time derivative. The calibration process is

to compare $ME$ with $RE$.

### 2.4 Automatized calculations algorithms

Used regression functions enforce the Levenberg-Marquardt algorithm for the $ER$ fits on the whole acquired interval and

singular-value decomposition (SVD) algorithm giving residuals for polynomial fits with the least square method for optimizing

fitting parameters. Then, the calculated polynomial roots are determined by a function based on a Riders algorithm. The first



root gives the starting point (first cross point between the real measurement curve and the *ER* regressed curve) and the third

root (third cross point) gives the limiting time for new *ER* fit.


## 3 Result and discussion

Soil evaporation measurement technique described in this paper is based on an adapted NSS technique principle. Used sensors characteristics, exact chamber configuration, regression calculations and wind influence are of a great importance.

### 3.1 Adaptation of dynamic closed NSS chamber

This technique is known since almost one century, described firstly by Bornemann in 1920 and used for measurements of a trace gas efflux such as $CO_2$, $N_2O$ or $CH_4$. Its operation principle is simple and consists to monitor the gas of interest concentration rise when a well delimited soil part (delimited by an inserted collar) is covered by a cloche (chamber head). Numerous variants of this technique are used and continually improved and described such as: respiration chambers: Pavelka

et al (2018), Open Top chambers: Fang and Moncrieff (1998) or forced diffusion chambers: Risk et al. (2011).

General issues concerning the closed chamber technique, construction, operations and wind corrections are listed elsewhere. In this paper, only special issues and solutions concerning notably the evaporation chamber ASERC technique corrected versus the wind are reported explicitly but the experimental setup also includes implicitly some general solutions adopted for the closed chamber technique.

NSS chambers technique needs to be carefully considered for the water evaporation measurement use, but when correctly used, it can provide valuable data. The usual assumption concerning this technique concerns the soil efflux which results from the gas of interest migration through the soil supposed to follow a pure molecular diffusion regime described by the Fick's laws. However, the evaporation efflux is a mass flow as during the evaporation, it means during the water to water-vapor transformation, the volume is strongly increased (about 1250-fold) which is moreover all vapor machines operation base. Then

a device allowing an additional gas to escape from the soil to the head but do not rising the internal pressure as well, needs to be implemented on the chamber. Different solutions are possible and actually concern all the closed chambers since the soil evaporation process is present even if the water vapor is not the gas of interest. One of the most used devices is a vent tube which gives rise to another problem: Venturi effect (Xu et al 2005, Bain et al. 2005, Suleau et al. 2009). Another solution consists of an expansion room implementation which allows to expand the chamber head volume maintaining the internal

pressure in equilibrium with the external air pressure. The expansion room is not subject to the Venturi effect. The late solution was then adopted for ASERC.





### 3.1.1 Sensors

The evaporation process is relatively fast and then, the humidity sensors should be even faster otherwise the deduced efflux may be biased. Fig. 02 (a) shows the accumulated effluxes measured by three sensors with a different response time. As we

can see, a different response time of different sensors biases results thereof. A simple simulation, Fig. 02 (b), of a signal given by a sensor with more or less slow response time along with an artificial start delay imposed by the operator (in the case of a leading pipes, we have to wait after the chamber deployment before to record air analysing data from a distant analyser) and shows a possible underestimation but also overestimation of deduced efflux. Such overestimation may be committed with a relatively fast (as P14 Rapid) but not very fast sensor (as FTUTA 34) and a long recording time. As the signal deformation

arises mainly at the beginning, during the recorded data regression, the first points importance will be overcome and the best fit will be accorded to the further points increasing artificially $ER$ amplitude and then the deduced flux. This possible overestimation vanishes with delay which results from the leading pipes or from a not instantaneous head space air mixing as shown later in this text. In the case of the leading pipe presence, the imposed time delay depends on the pipe's diameter and the air flow value. Flowing air is always mixed with pipe's enclosed air making calculated time delay rather approximative

and not very well defined. An embedded sensor is always preferable. The fastest reliable air moisture and air temperature sensor used is a P14 Rapid (response time $\tau_{63} < 1.5s$) on a Linpicco plate holding a PT1000 sensor providing simultaneous humidity RH (%) and air temperature Ta (°C) measurement.





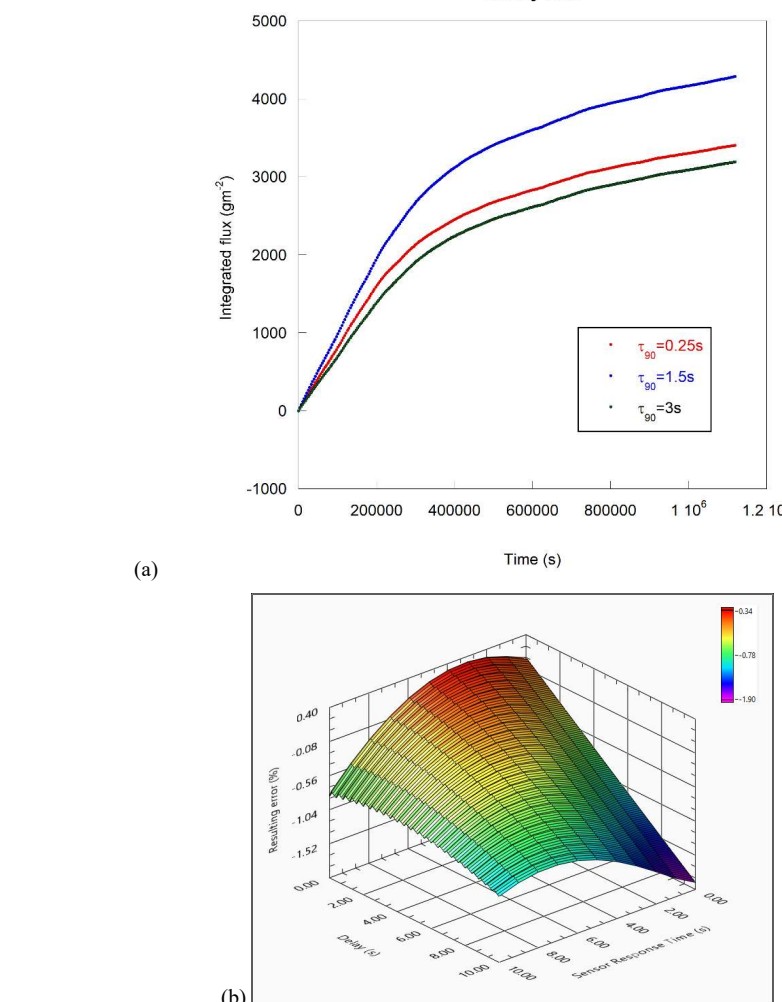

245        (a)

(b)

**Figure 02: (a) Integrate *ME* given by tree different sensor with different response time. BME280 and 3s response time, P14 Rapid with 1.5s response time and F-Tuta 34 which has the faster response time of 0.25s, unfortunately this sensor present quickly malfunctions and then was discarded. (b) Simulation of the *ER* regression error due to a slow sensor and introduced waiting delay with origin of time change.**






A simple calculation based on an empirical water saturation pressure versus temperature law published by Wagner (1995) gives the absolute humidity $q$ (g/m³). This formula is accurate to within 0.1% over the temperature range –30°C to +35°C:

$$q = \frac{13.2471488 \times e^{\frac{17.67 \times Ta}{243.5 + Ta}} \times RH}{273.15 + Ta}$$


(6)

An external sensor such as Infra-Red Gas Analyzer (IRGA) provides better accuracy but requires a leading pipe use between the chamber head and the IRGA with an external pump. The leading pipes may seriously bias the measures by adsorption problems, condensation problems and time lag between the chamber closure and the corresponding air sample measure, additionally to the heating problem since the IRGA is heating the analysed air sample which is reinjected back to the chamber
head and needs then to be cool down.

The usual polyurethane (PU) pneumatics tubes were checked. Fig. 03 (a) shows an apparent characteristic time variation of a measured absolute humidity rise by a fast IRGA (Li-840A, LI-COR Biosciences, Lincoln, Nebraska USA) on one edge of the leading pipes when a step like humidity rise or fall is induced (by a Li-610 portable dew point generator, LI-COR Biosciences, Lincoln, Nebraska USA) plugging or unplugging on the other edge of the leading pipes versus the length of the pipes. As we
can see, the apparent characteristic time is increasing strongly with the leading pipes length reflecting a strong sorption problem with the PU pipes. Polytetrafluoroethylene (PTFE also called Teflon) pipes are preferable but need to be insulated anyway in order to prevent condensation possibility and induce always a lag problem between the chamber closure time and the incoming air sample time from the chamber to the analyser which needs to be precisely known as it may bias again the regression results. An embedded fast and accurate sensor is then preferable.




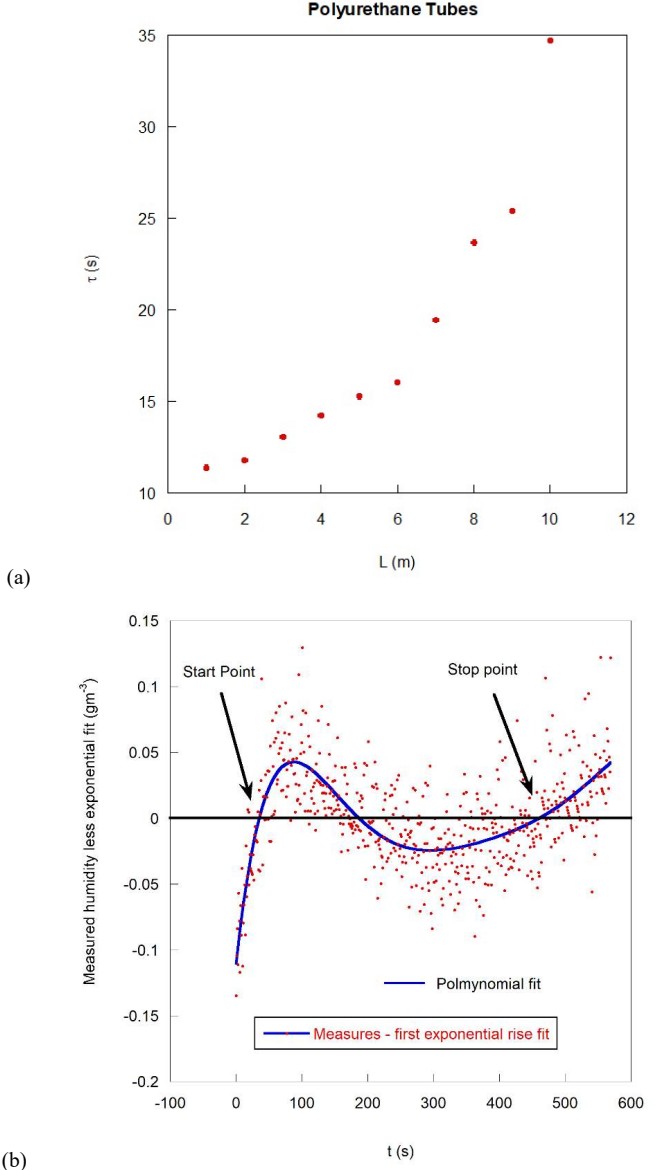

(a)

(b)

**Figure 03: (a) Apparent response time of the polyurethane tubes versus the length of the tubes. (b) Measured absolute humidity less exponential fit (residuals) giving start and stop points for a second regression**.




### 3.1.2 Regressions calculation methodology

As in the case of all other soil trace gases, the measured absolute water vapor concentration rise in a closed chamber head is not linear with time but rather follows an Exponential Rise (ER) law. As usually, due to the complexity of the accumulation

and feedback process, the exponential law does not describe perfectly the measurements and some deviations are observed making the regression results sensitive to the starting point, duration and the end point. This is a general observation intrinsically tied with the closed chamber technique (Nakano et al. 2004). Concerning water evaporation, we can notice that the measurement curve and the ER fit for a long enough time present three well defined cross points. For this study, in order to stabilize the numerical regression conditions (fit), the starting point is systematically chosen at the first cross point and the

end point at the third cross point (Fig. 03 (b)). In other words, a double fit is needed. A first fit on the whole disponible length provides the starting and the end point and then, a second exponential rise fit is performed between these two points provide every seeking values. Only the result of the second fit is considered as reliable. Of course, in some cases such as a very slow evaporation, we do not observe the third cross point. In these cases, the retained interval is between the first cross point and the last available point.

Once the experimental setup truly built, the operation condition chosen and the regression points stabilized, the wind influence was studied.

### 3.1.3 Wind influence considerations

The wind influence on the water vapor efflux is well known and widely studied (Thornthwaite and Holzman 1942, Hanks and

Woodruff 1958). Even if a non-diffusive regime for a soil evaporation process was explored more than a half century ago (Fukuda 1955), studies which considered a gustiness wind influence from the theoretical point of view by a sinusoidal representation and concluded to a negligibility of the phenomenon, others authors studied and experimented the wind influenced evaporation (Farrell et al. 1966, Scotter and Raats 1968) and concluded, on the contrary, to a great importance of a non-diffusive regime. Recently, numerous authors, experimentally and theoretically studding the gas propagation in the porous

media, have pointed out an important or even biggest role of the non-diffusive regimes such as thermal and solutal dispersion (Davarzani et al. 2014), convection and advection or pressure fluctuations for gas movement through porous media (soil) see Sánchez-Cañete et al. (2016) and the references given by them. One of a major gas movement cause is the so-called wind pumping which includes three effects:



- Venturi effect (Xu et al. 2005, Bain et al. 2005, Suleau et al. 2009) giving rise to a mass transfer by pressure gradient
establishment.
- Natural gradient concentrations disturbance (Le Dantec et al. 1999, Longdoz et al. 2000, Lai et al. 2012) playing an important role during very calm conditions and a highly stratified boundary layer slowing down the diffusion efflux and, once disturbed, by the head space fan mixing releasing an unusual high apparent efflux.
- Eddy pressure fluctuations causing gas dispersion (Maier et al.2010, Mohr et al. 2016, Brændholt et al. 2017,
Pourbakhtiar et al. 2017, Poulsen et al. 2017, Mohr et al. 2017) that may be of a very important gas transport regime. This effect is more or less screened by the chamber deployment mainly depending on the wind importance versus the head space mixing fan disturbances.

All these effects are altered by a deployed chamber head and then affect the closed chamber measurements versus a natural
soil efflux. As the wind cannot be reproduced inside the chamber heads, the only possibility is to correct the data calibrating the measurements versus the wind which is the aim of this paper presenting a non-steady state dynamic chamber technique adapted and wind corrected for the water vapor efflux measurement.

### 3.1.4 Modeling difficulties

The wind influence on the soil evaporation depends on numerous variables such as the soil temperature and moisture, the air temperature and the humidity but also the soil nature and the soil texture. These variables may change more or less quickly and some of them such as the soil texture are not real-time monitorable.

In others words, even if we succeed to model the wind effect, we will not be able to use it for chambers data corrections. An auto-calibrating chamber would be considered as a solution. The target is not to measure every variable and inject it into a
complex model but rather to measure the "susceptibility" of the soil evaporation to the wind and correct the chamber measurements against the measured wind. The following protocol gives very simples yet relatively accurate results.

### 3.1.5 Fundamental finding

At a fixed internal fan speed (PWM constant), the Measured water vapor Efflux ($ME$) is not directly proportional to the Real
water vapor Efflux ($RE$) during a soil drying process at a stable wind (Fig. 04).

This non-linearity changes with the soil nature and even the soil texture is making impossible to correct any data for all soil natures and textures with a unique formula based only on the wind speed.



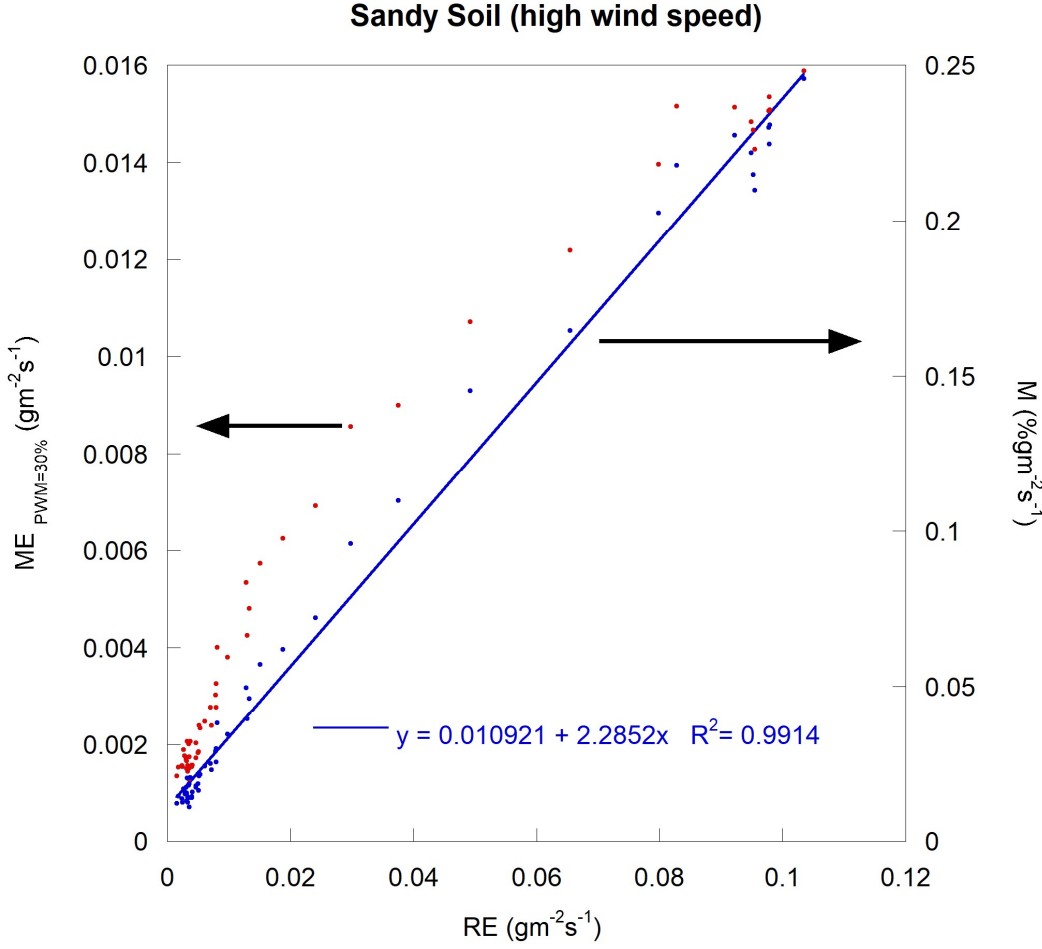

**Figure 04:** Measure evaporation versus real evaporation (left side axis) and M, averaged measure multiplied by Z
factor, along with a linear regression on the right-side axis for sandy soil under 1.15ms$^{-1}$ wind.





Geoscientific Instrumentation
### 3.1.6 Embedded fan influence

As we can see, in the adopted configuration, the fan influence is similar to the wind influence. Similar, but not identical since the wind brings some fresh air when a fan can only mix the internal head space air with a progressively rising water vapor

concentration. Figure 05 shows also the recorded water vapor efflux with a sandy and a clay soil versus the fan PWM duty control. As we can see, both soils results are very well described by an exponential law:

$$ME(PWM) = A * e^{-Z/PWM}$$

(7)

$A$ and $Z$ being constant for a given soil sample and external conditions.

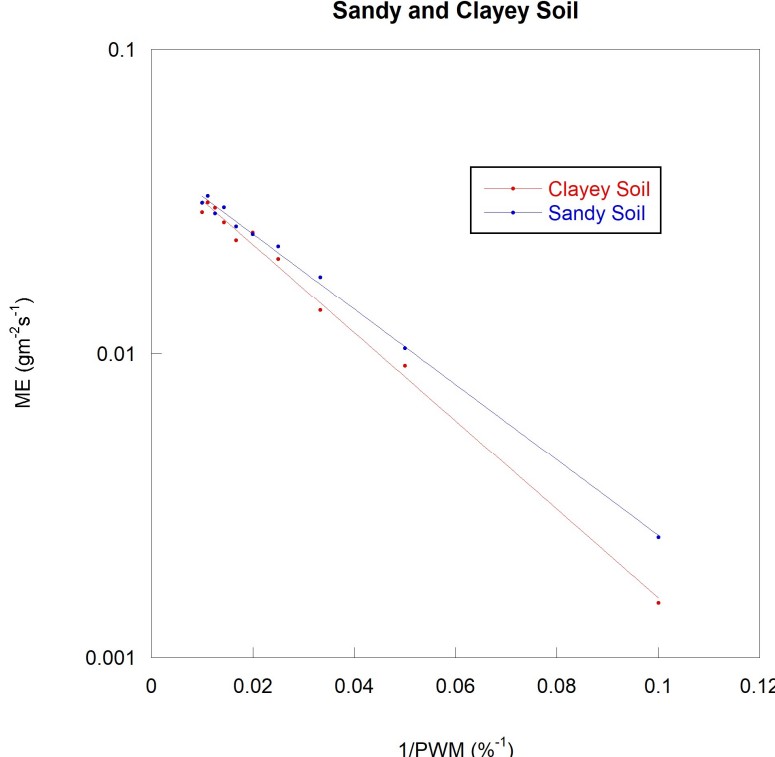


**Figure 05: Measured evaporation versus 1/PWM for sandy soil under 0.72ms⁻¹ wind and for clayey soil under 0.8ms⁻¹ wind.**



The constant $A$ is reflecting the amplitude of the evaporation for a given soil and weather conditions and the constant $Z$ is

reflecting a soil susceptibility to the internal fan mixing flow. By similitude, one can assume $Z$ reflecting the soil evaporation

wind susceptibility.

In order to determine $Z$, the most effective way is to perform numerous measurements with a different $PWM$ value and apply

an exponential regression to all these results ($Z_{total}$). However, this way is long and resulting perturbations do not allow a high

sampling rate. Two-point measurements used for $Z$ determination are relatively of a good concordance if the first fan speed

used is low such as $PWM = 10\%$ and the second speed is significantly higher such as PWM = 30% (Fig. 06).

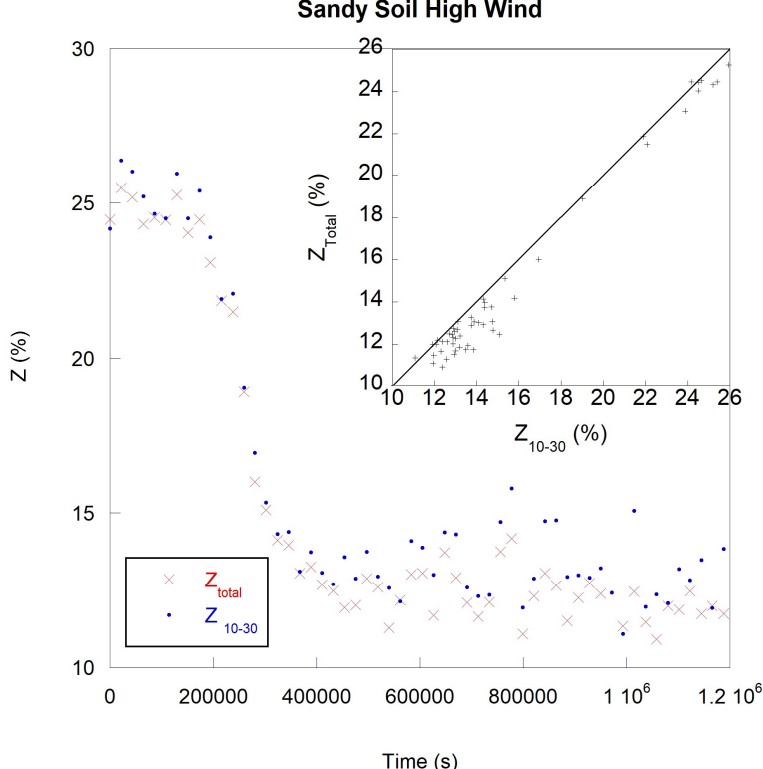

**Figure 06: $Z_{total}$ and $Z_{10-30}$ versus time for sandy soil at relatively strong wind of 1.15ms$^{-1}$. Insert, the same values of $Z_{total}$ versus $Z_{10-30}$.**






The best results for a correction were given by the function $M$ which is the average of $ME_{10}$ at $PWM = 10\%$ and $ME_{30}$ at $PWM = 30\%$ multiplied by $Z_{10\text{-}30}$ obtained by an exponential regression of the two-point measurement $ME_{10}$ and $ME_{30}$ versus $1/PWM$ by an exponential function 6:

$$M = \frac{ME_{10} + ME_{30}}{2} * Z_{10-3}$$

365                                                                                                                                (8)

As we can see on the Fig. 04, the $RE$ is near proportional to $M$ (defined by formula 7) and, which is the main benefit of $Z$ introduction, this proportionality constant $m$ depends mainly on the wind speed $w_S$ and remains unchanged whatever is the soil nature, texture, moisture or temperature as shows Fig. 07. Only M depends on the soil nature, texture or soil and air meteoritical variables what is included in the measurements.

$$RE = m(w_S) * M + B$$

                                                                                                                                    (9)

$B$ is a constant, of a very small amplitude useful only for a very small wind speed or a very dry soil.

The $m$ dependence of the wind speed is not trivial. The plateau around $w = 0.5\text{ms}^{-1}$ corresponds probably to the fan perturbations at $PWM = 20\%$ (an average between PWM=10% and PWM=30%) comparable to the wind of $0.5\text{ms}^{-1}$ speed

perturbations. This particular value is tied with the chamber design and cannot be used as a universal value. The other limitation is the high wind speed. As shows Smits and al. (2015), for the wind speed superior to a threshold value $w_{Smax}$, the evaporation process is not much more affected by the wind. Then, $m(w)$ is probably no more linear with $w_S > w_{Smax}$.

For the studied range of the wind speed, the adopted adjustment formula for $m(w_S)$ is of the form:

$$m(w_S) = \frac{a * w_S}{e^{b/w_S}} + c(1 - e^{-d*w_S}) + g$$

380                                                                                                                                (10)

With a, b, c, d and g constants determined empirically.

As a validity check for the studied wind speed in the range of zero to $2\text{ms}^{-1}$, Fig. 08 shows all the available data corrected using $m(w_S)$ the unique function for all soils' nature and texture depending only on the wind speed. The linear regression shows a

reasonable concordance with $R^2 = 0.95$.



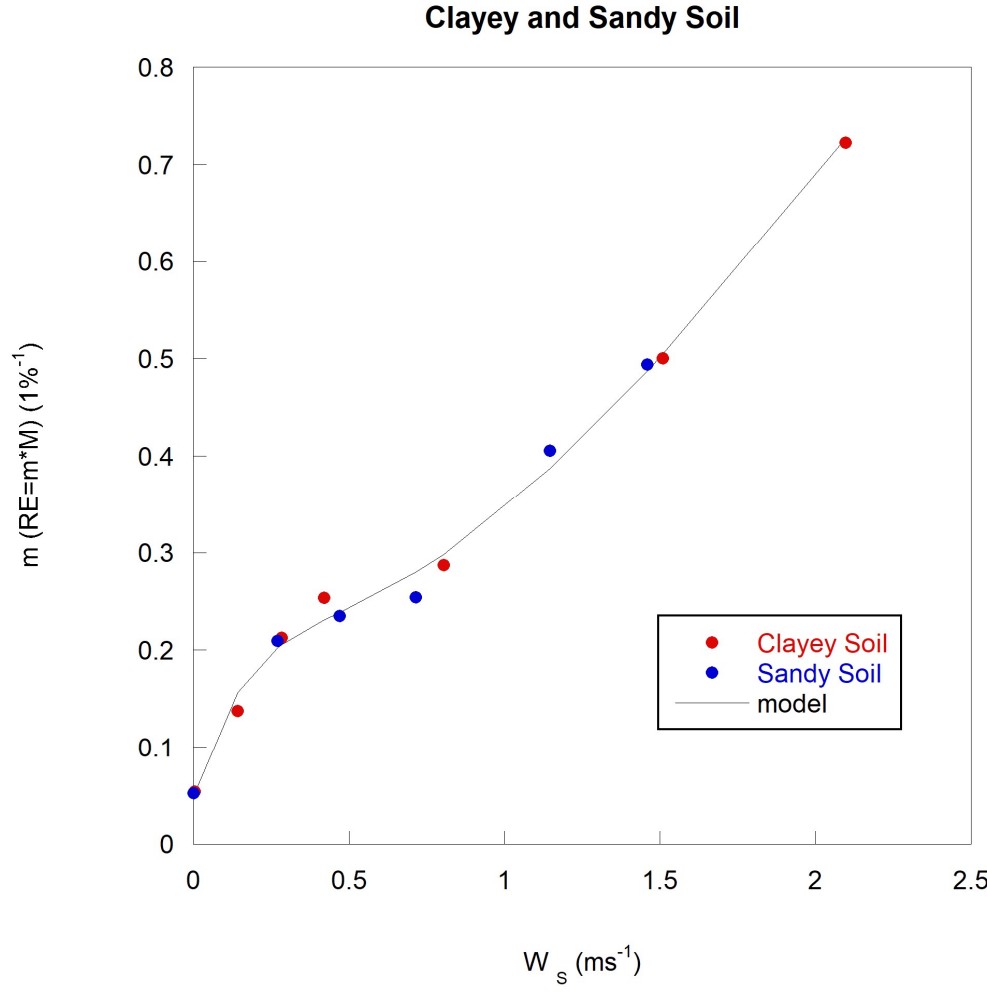

**Figure 07:** *m* versus gravimetric soil water content₅ for sandy soil and clayey soil.



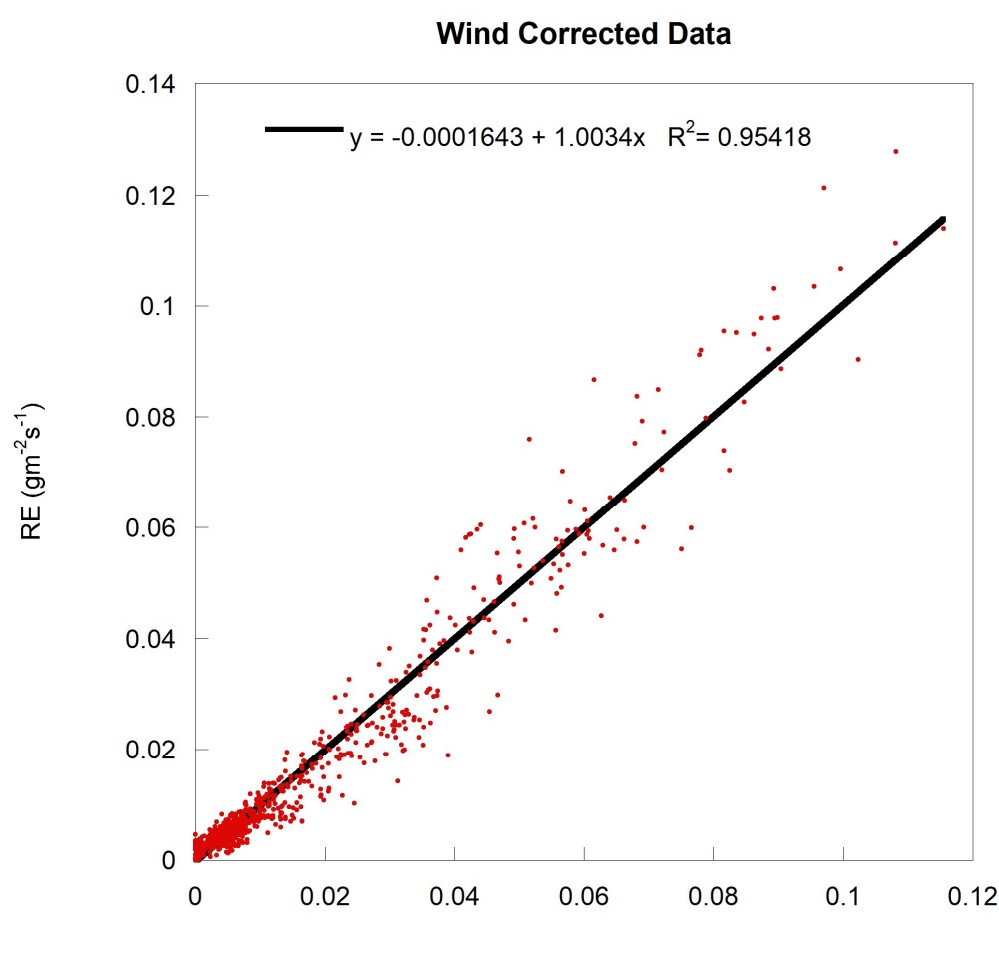


**Figure 08: Real evaporation versus m for all disponible data along with a linear regression.**




### 3.2 Laboratory measured soil evaporation *ME* results

It is well known and admitted that the soil evaporation can be divided into three stages. (Introduced by Philip 1959, Wilsdon and al. 1994 or Hiller 2004). The wet soil, water saturated or near to water saturated, is evaporating with a constant rate greatly depending on the wind. With the progressively drying soil, a second stage appears after the so-called Air Entry Value (AEV) and shows a smallest wind dependence. A third stage is concerning the very dry soils with a constant extremely low evaporation rate and was not really observed in this study except the zero-wind sand evaporation record that took over two months of

constant monitoring. In order to compare measurements under different wind speeds, such as the real evaporation from sandy and from clayey soils, a semi logarithmic scale versus soil moisture is probably the most relevant way Fig. 09. On these figures, we can notice that regarding the sand, the first stage is important comparatively to the clayey soil where the evaporation is quickly falling to the second stage. This behaviour is characteristic for a relatively low-rate evaporation on a sandy soil (Holmes 1961). The second stage displays a very linear behaviour for the sand and also for the clay on the logarithmic scale. This means

that, in the second stage, the evaporation rate is as an exponential function of the soil moisture *w*.

$$RE_{Second\ Stage} = C(w_S) * e^{D*w}$$

(11)

With *C* depending slightly on the wind speed, $w_S$ and *D* being a constant almost independent of the wind for a sandy soil.

Indeed, regarding the sandy soil, whatever is the wind, the slopes are the same and the curves are parallel but remain slightly wind affected since they are not superimposed.

For the clayey soil, the second stage evaporation rates are higher for the higher wind. There is a well visible common point where the evaporation rates are the same whatever is the wind. For a lower moisture, the curves clearly diverge as the slopes are different.

The difference between a sandy soil and a clayey soil draying process is certainly affected by the micro and later macro desiccation crack apparitions in the clayey soils (Lau 1987, Morris et al. 1992, Kodikara 2002). These cracks may be considered as an effective soil/air interface surface increase and then an additive water vapor exchange which may significantly increase the evaporation rate under the wind (Nachshon et al. 2012). The other difference between theses soils is the grain seize difference and then the intergrain void space, and the resulting matric suction amplitude as discussed later in this paper.





420          (a)

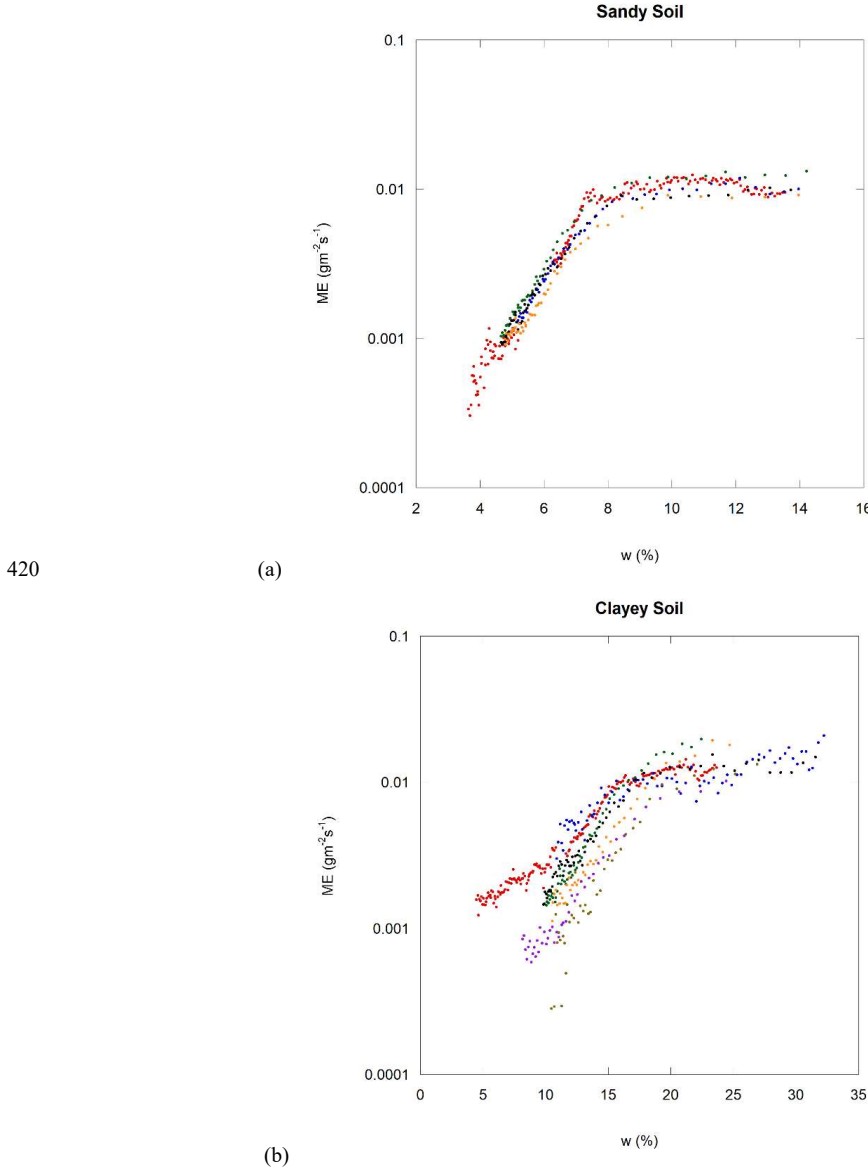

(b)

**Figure 10: Measured Evaporation *ME* versus gravimetric soil moisture (PWM=30%); (a) for sandy soil and (b) for clayey soil for several wind speeds.**



### 3.2.1 Wind affected measures *ME*


Figure 10 resumes *ME* versus the soil moisture for a sandy soil (a) and a clayey soil (b). For the sandy soil, as we can expect, since the measures are done under a chamber cloche that isolates the soil from the wind, the measures do not display a clear wind dependence. On the contrary, for a clayey soil, conversely to the *RE*, *ME* decreases with the increasing wind speed. This finding may be explained by two facts:


Comparatively to the trace gas efflux such as $CO_2$, $CH_4$ or $N_2O$ effluxes, the water vapor efflux *RE* has two sources-sinks/components: Production (*P*) and Stock (*S*) from the soil pores, by dissolution into the soil water or by sorption (absorption plus adsorption).

$$RE = P - \frac{\partial S}{\partial t}$$

(12)


Real surface efflux is then a result of the production less stock variation.

The wind may have a great influence on the efflux by forcing to unstock but much less or ever nil influence on the production itself in the case of trace gases such as $CO_2$, $N_2O$ or $CH_4$ and the deep subsurface evaporation with a low porosity soil.

In the case of a water vapor efflux, the soil water vapor stocking ability exists as well in the case of a Dry Soil Layer formation (DSL) (E. Balugani et al., 2018) that concerns mainly the sandy soils under arid or semi-arid climate (Wang 2015), as in the


case of a simply non-saturated soil (vadose zone) (Balugani et al. 2016), in both cases the stockage is realized in the soil pores saturated with the water vapor or by soil (mainly clay) sorption. The migration of the water vapor from this undersurface zone is predominant *RE* process in the evaporation second stage (Geistlinger et al., 2018). Moreover, on one hand, concerning the evaporation, the wind may influence directly the production in the shallow subsurface (Harris 1916, Smits 2015, a quasi-exhaustive list of the evaporations factors is given by Faseel Suleman et al., 2017) and, on the other hand, the soil is able to


absorb the water vapor from a deeper and wetter evaporating soil layer or from the air making the stock *S* dependent not only on the soil water vapor production *P* but also on the soil/atmosphere surface interaction (Amer 2015). This last point may explain why the sandy soil chamber-based measurements are independent of the external wind when the clayey soil gives the measurements of the water vapor effluxes decreasing with the increasing external wind.

Indeed, the soil ability to absorb water vapor is increasing with its particles size decreasing (Chiorean 2017) or, by definition,


the sand particles are of a several magnitude bigger size than the clay particles. The sandy soil is then much less able to absorb water vapor comparatively to the clayey soil which the water vapor sorption is well known and documented from an experimental and theoretical point of view (Johansen and Dunning 1957, Likos and Lu 2002, Leelamanie 2010 or Arthur et al. 2016). In the windy conditions, the soil moisture top layers moisture is an equilibrium between the wind pumping and the soil water absorption/retention forces. When the wind cease, another equilibrium has to be reached with a down soil layer of higher



moisture. A short-term water vapor sorption by the previously wind dried soil layer may significantly temporarily lower down
the apparent evaporation rate $RE$ (Jabro 2009).

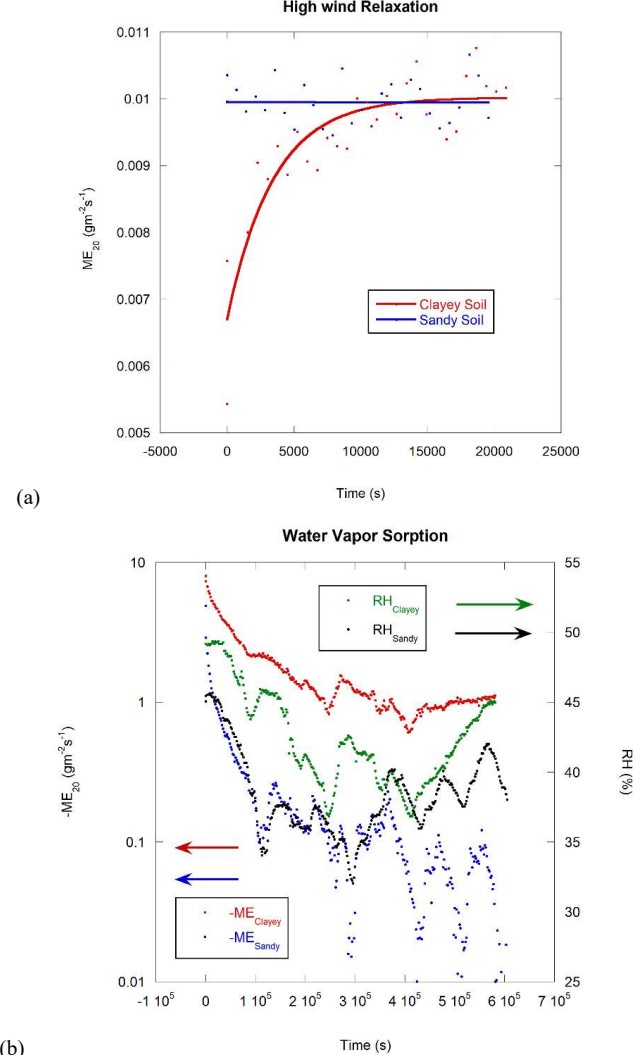

(a)

(b)

**Figure 11: Measured (at PWM=20%) evaporation $ME$ versus time (a) after a strong wind ($W_S$ = 2ms$^{-1}$) for clayey and**
**sandy soil. Slid line represents a linear regression for sandy soil and an exponential regression for clayey soil. (b) after**
**oven drying for 24H at 105°C.**





Figure 11 (a) shows *ME* behaviour at always the same fan speed (*PWM* = 20%) over a wet sandy or clayey soil after one day
of a strong wind that ceases immediately after the first measurement engagement (30 measurements, each chamber deployment
during 10 minutes inters spaced by chamber opening and head space flushing during 1 minute).

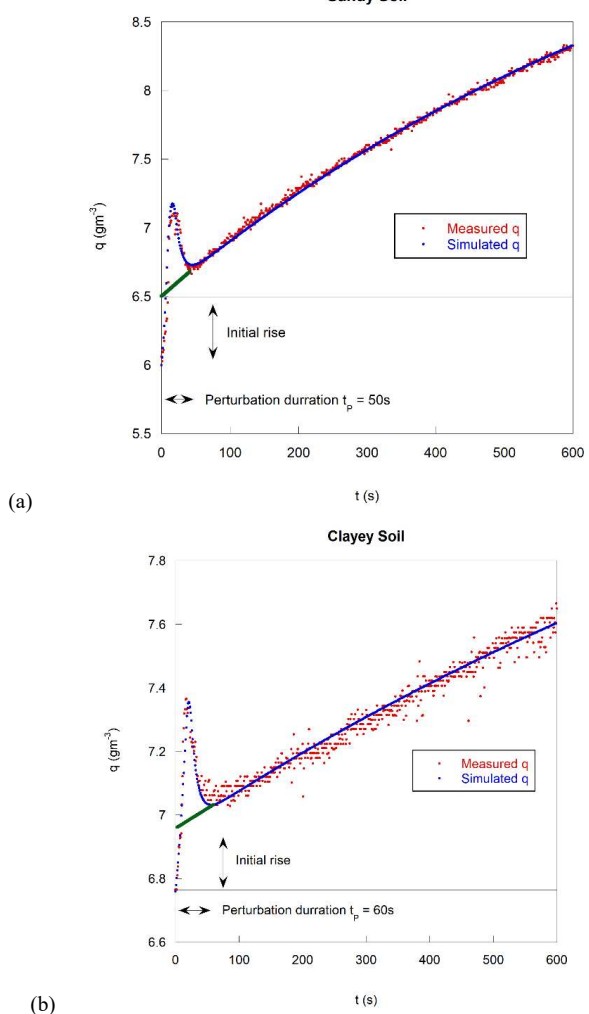

(a)

(b)

**Figure 12: Absolute water vapor concentration versus time. Measured concentration and simulated concentration, (a)
sandy soil and (b) clayey soil.**






### 3.2.1.1 Inertia and chamber head air mixing time

During a chamber head deployment with a high-speed wind and a relatively wet sandy soil, an initial peak is observed in the
enclosed air absolute humidity curve versus time (Fig. 12(a) for sandy soil and 12 (b) for clayey soil).
This is a direct consequence of a non-instantaneous head space air mixing coupled with a high-water vapor efflux in the
boundary layer over the soil forced by the wind and is qualitatively well described by a very simple model of a mixed closed
space air. A bigger volume of a low humidity air mixed with a smallest volume of a higher humidity air (boundary layer
volume) intaking linearly is increasing humidity reaching a maximum and then is rapidly decreasing reaching a usual *ER*
evolution (see the scheme on Fig. 13(a)). This mixing process may be described by an equation of the form:
$$C(t + dt) = a * C(t) + (1 - a) * S(t)$$

(13)

With C(t) being the measured concentration of interest, a being the mixing ratio defined by the proportion of the recycled air
divided by the proportion of the air coming from the boundary layer and S(t) being concentration within this layer. S(t) is a
sum of a usual ER evolution $S_{ER}(t) = C_L-(C_L-C_S)*exp(-t/\tau_0)$ with an overage of the water vapor due to the residue of a high
wind forced efflux $S_R(t)$. This overage concentration is modelized starting from zero, reaching linearly a maximum value
during about a half of the perturbation delay $t_P$ and then decreasing quickly always linearly with time (Fig. 13(b)).

It reflects a boundary air layer of a high moisture, that is formed by a strong wind forced water vapor efflux, which is not
immediately stopped by the chamber head closure (residual efflux). The resulting perturbation ceases after 50 seconds of the
chamber head deployment for a sandy soil. This initial peak vanishes with a lower wind speed or a lower soil humidity or also
a higher fan speed. For example, under PWM = 20%, the initial peak is hard to spot and is no more visible with the higher fan
speeds whatever is the wind speed (in the studied range) or the soil humidity. The peak vanishes but an initial quick humidity
rise inside the deployed chamber head is still visible.


Figure 12 displays, measured along a simulation, the resulting absolute water concentration inside the chamber head; (a) for a
sandy soil and (b) for a clayey soil. Figure 13 provide the adjustment constants definitions. Additionally, we can understand:
- $C_0$ as a starting concentration within the chamber head air (ambient concentration);
- $C_1$ as the maximum concentration within the boundary layer due to the residual inertial efflux enrichment in competition with
the chamber head enclosing the air mixing by the embedded fan;
- $C_S$ as the concentration resulting from the initial concentration raised by the residual inertial efflux ($S_R$ is the residual surplus
of the boundary concentration due only to the residual inertial contribution), a is the fan mixing ratio, $t_m$ is the residual efflux





duration, $t_e$ time of the residual efflux duration enhanced by the effective mixing time (13s in our case), $\tau_0$ resulting characteristic time for an *ER* concentration evolution; and

- $C_L$ is the concentration of the boundary layer but on the soil side (limit of the concentration within the deployed chamber head after an infinite duration).

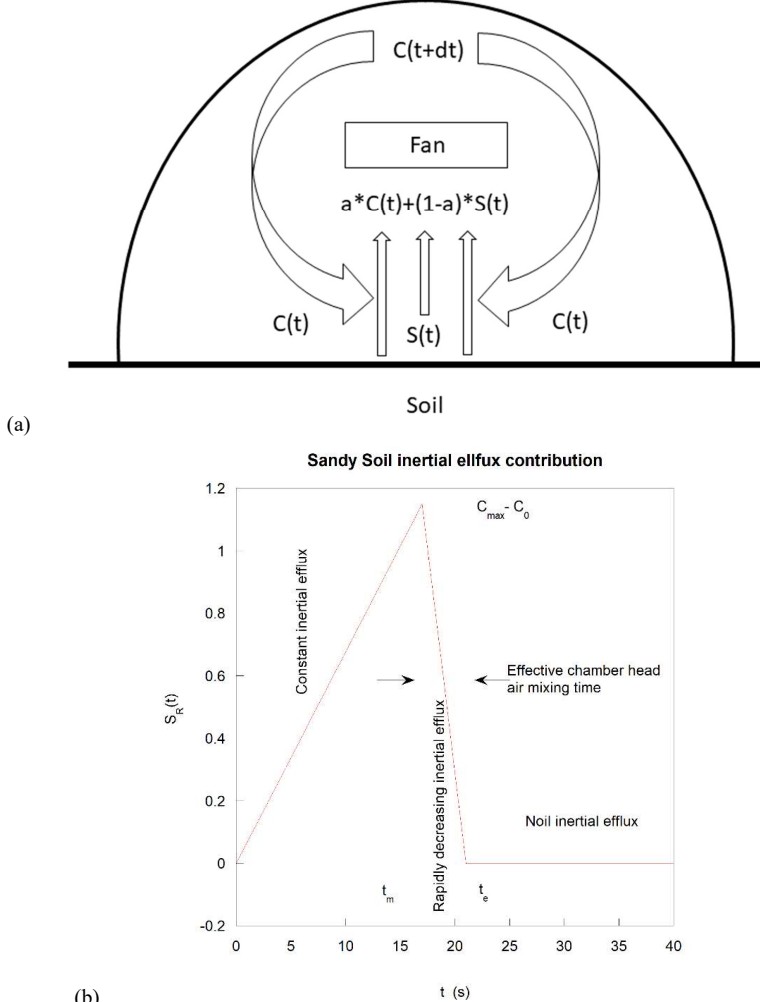

(a)

(b)

**Figure 13: (a) Chamber head air mixing principle. (b) Residual water vapor efflux to the boundary layer water vapor**

**concentration $S_R$ versus time.**



Indeed, the same measurement (PWM=10% under a high wind and a soil moisture) performed with a clayey soil (Fig. 12(b)) shows also an initial peak but, comparatively to the sandy soil (Fig. 12(a)), this peak, as the whole evaporation rate, is of a

smallest amplitude since the water vapor efflux, as discussed previously, is absorbed by the top soil layer initially dried by the wind pumping. With a clayey soil, the initial peak perturbation duration is also slightly longer (60s) comparatively to a sandy soil (50s).

This initial peak requires a special attention during the *ER* regression calculations in order to do not bias the results. An initial short time laps exclusion, 50s-60s in this case, may be necessary. The data points should be discarded but it is important to do

not change the time origin because it may lead to an important flux calculation bias (5% in this sandy soil case). The amount of the water vapor released during first 50s is rising significantly the initial water vapor concentration measurements and the total water vapor content inside the chamber head (initial rise), but does not impact further efflux calculations.

For both simulations, the constants tied with the chamber design such as $a = 0.9$ and effective mixing time $t_e - t_m = 13s$, are the

same and only constants tied with the soil samples nature change.

Sandy soil: $C_0 = 6$ g/m³, $C_I = 1.35$ g/m³, $C_S = 6.55$ g/m³, $C_L = 9.7$ g/m³, $t_e = 21s$, $t_m = 8s$, $\tau_0 = 900s$.

Clayey soil: $C_0 = 6.76$ g/m³, $C_I = 0.8$ g/m³, $C_S = 6.96$ g/m³, $C_L = 8.74$ g/m³, $t_e = 30s$, $t_m = 17s$, $\tau_0 = 1500s$.

Comparatively to the sandy soil, the residual efflux due to the wind is of a smallest amplitude but longer.

### 3.2.1.2 ME: results for sandy soil

The wind is draying the soil sample but the chamber deployment, even if it insulates the soil sample from the external wind, does not give an opportunity to the shallow, wind dried, layer of a sandy soil to reabsorb water vapor from the air or from the deeper soil, and does not limit the measured evaporation rate (figure 10). It is worth to be noted that for both, the sandy soil and the clayey soil, the difference of the real evaporation is about one decade between a zero-wind evaporation and a small wind evaporation. This can be attributed to a shallow boundary air layer over the soil that is disturbed by any wind. Without

the wind, this boundary has a high-water vapor content, limiting the evaporation from the soil by molecular diffusion. A slow advection is also present (water vapor is lighter than the air) but this transport is visibly very slow.

### 3.2.1.3 ME: results for clayey soil

The wind is draying the soil sample in the same way as for the sandy soil. However, after the chamber deployment, a previously wind drayed clayey soil layer is absorbing the water vapor from the deeper wetter soil reducing the net water vapor efflux *ME*.

The measured surface efflux is real but the conditions are not. We are in a transition regime caused by the chamber deployment



and the measured soil portion isolation from the external wind. In the sandy soil case, this transition period is very short (*ME* is wind independent) and does not affect the measurement when, in the case of a clayey soil, this transition period is long and the characteristic time deduced from an exponential rise regression is about one hour long. The clayey soil sample needs about four hours to reach equilibrium from under a wind speed (*Ws*) $Ws = 2\text{ms}^{-1}$ to new equilibrium under a $Ws \approx 0\text{ms}^{-1}$ wind.

Figure 11 (b) displays the *ME* behaviour at PWM=20% but this time, the soil sample was oven dried for 24H at 105°C. The bucket with dried soil samples were sealed and opened just before the first measurement. As we can see, both, the clayey soil and the sandy soil are sorbing the water vapor from the atmosphere (negative *ME*) but the sandy soil sorption is quickly falling to nearly zero when the clayey soil sorption is perduring and is inversely proportional to the air water demand.







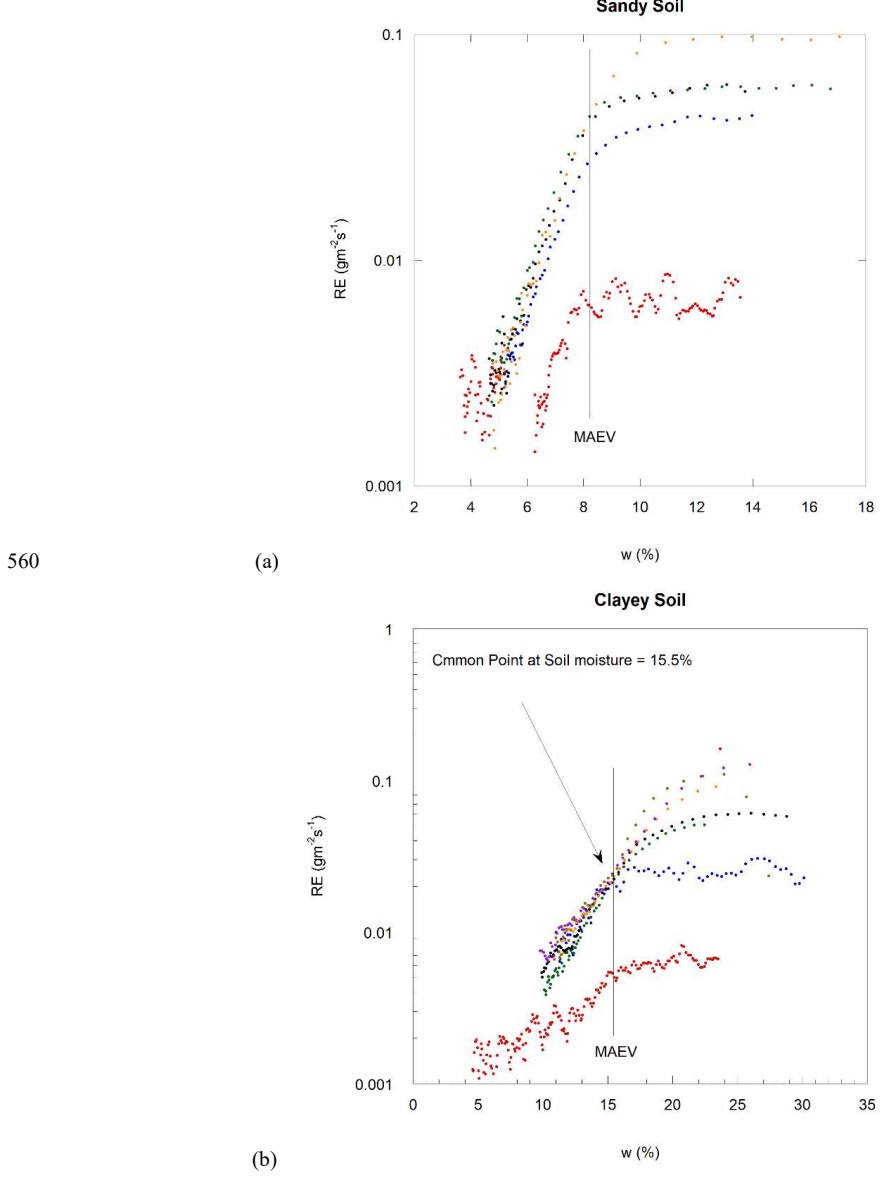

560        (a)

(b)

**Figure 09: Real evaporation versus gravimetric soil moisture a) for sandy soil and b) for clayey soil for several wind speeds.**





**3.2.2 Real Evaporation *RE* of clayey soil**

**3.2.2.1 Soil cracking**

The cracks formation in a drying clayey soil, also called desiccation soil cracks, has a great influence on the total evaporation up to 50% enhancement according to Hatano et al. (1988) and, under the windy condition, the total evaporation may be even increased by two orders of magnitude (Nachshon et all. 2012). This phenomenon is widely studied and relatively well

documented as its consequences for the engineering (Lytton et al. 1976, Daniel et al 1993, Kodikara et al. 2002, Rodriguez et al. 2007, Stirling et al. 2017) and the agriculture (Pal et al. 2012, Kurtzman et al. 2016) are very important. Picture 14 shows the clayey soil sample in its dry state; (a) dried under a moderate wind, (b) dried under no wind. During this study, different cracks patterns are obtained with a different wind speed and a different drying ratio. An obvious wind importance on the cracking pattern was noticed. Higher is the wind more numerous are the cracks in accordance with existing studies (Corte and

Higashi 1960, Tang et al. 2008 and 2010, Costa et al. 2013) and coarsest are the resulting cracks pattern always in accordance with the previous studies (Corte and Higashi 1960, Lau 1987, Kodikara et al. 2000, Nahlawi and Kodikara 2006, Tang et al. 2008 and 2010, Costa et al. 2013), due to an important matric suction increase with the drying ratio in a clayey soil. This is a part of so-called dynamic effects. The studied clayey soil sample can lose up to 15% of its initial volume and its observation drying under different winds speeds allows to point out an interesting finding which is an existence of a common point (*CP*)

and a change in the evaporation ratio versus soil moisture slope below this point under all wind speeds but nil. The clayey soil sample is cracking under the wind but is drying as a whole block without cracks under the nil wind (Fig. 12(b)) creating a void space between the soil block and the bucket wall.

*CP* corresponds to a well-defined and constant *RE* about 0.024 gm⁻²s at a constant soil moisture *w* about 15.5%. It corresponds also to the *RE* versus *w* slope change for a drying soil. At the current research advancement point, one can only propose a

hypothesis to explain this phenomenon tied with the clayey soil Matrix air Entry Value (MAEV) corresponding to the air seepage through the soil matrix, but the present study does not allow to prove it. Soil *RE* is slowing down with the soil moisture decrease, however, in the swelling clayey soil case, the soil moisture decreases cause also micro-cracks and hollows formation, depending on the expansive clay content (smectite minerals, including montmorillonite and bentonite) and more precisely its vertisol character (Ahamad 1996, Everest et al. 2016), increasing effective soil/atmosphere interaction surface (Bronswijk

590 1988).





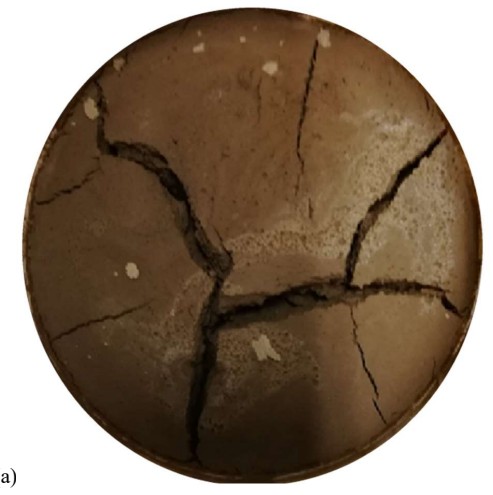

(a)

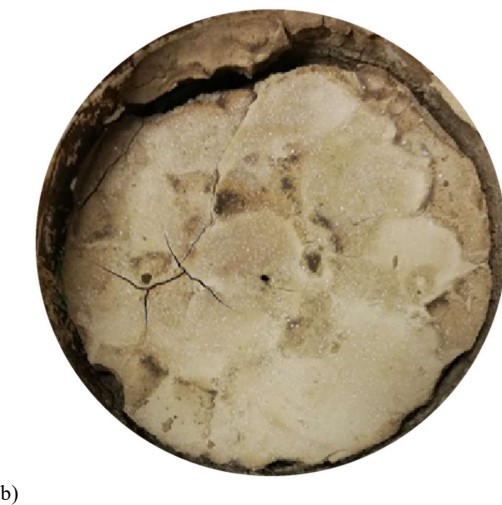

(b)

**Figure 14: Dried clayey soil surface after drying under (a) moderate wind of 0.8ms⁻¹ (b) no wind.**


This fact agrees with a relatively less pronounced transition between the first and the second stage of the evaporation observed in a clayey soil versus a sandy soil (Fig. 09). As the wind influence importance on the *RE* goes together with the interaction



surface, any change in the later is visible in the former. The cracks formation in the soil would increase *RE* changing the *RE*
versus *w* ratio and may then explain the apparent slope change observed in the Fig. 09 (b). However, the cracks transform the

soil sample to a fractured media, and are visible well before the *CP* point as showed by Song et al. 2016, cracking are forming
a so-called Cracking Air Entry Value (CAEV) on a moistest, almost saturated, clayey soil and their formation is progressive
which is rather incompatible with a brusque slope change. The concerned soil moisture point *CP* seems to correspond rather
to a soil Matrix Air Entry Value (Azam et al. 2013) below which the soil is acting as a porous media. The crack formation
causing the first AEV (CAEV) on a soil moisture characteristic curve is followed by the matrix AEV (MAEV) formation on a

drier soil and is very similar to a bimodal grain-size distribution curve (Satyanaga et al. 2013) where both grains of big and
small sizes are present in the soil with then two intergrains void space sizes. These two points: cracks and matrix AEV, happen
in a draying clayey under the wind before the final Void Ratio stabilization (Péron and Laloui 2005) and affect an evaporation
ratio which is one order of magnitude bigger than those of no wind evaporation. Since, under no wind, the CP is not visible,
one can deduce that the cracks or/and the wind presence is necessary for it.


**Conclusion.** The aim of these studies was to build a self-calibrating chamber based on an NSS technique and a simple working
protocol to correct the measured data versus the wind. The proposed chamber design along with a correction protocol allow a
$R^2 = 0.95$ confidence on a sandy or clayey soil and the surface wind in the range of 0 to 2ms$^{-1}$. The correction function has
only one variable; the wind speed (few centimetres above the soil surface) regardless of any other parameter such as soil nature,

soil texture, soil temperature or meteorological variable. However, a study of the higher wind speeds is suitable but exceeds
the ability of the present experimental set up. The presented results are valid for a bare soil chamber-based measurement. In
the vegetated plot case, the measured wind speed on the chamber level will be comparatively slow, however, the wind influence
is still important and forced below the canopy wind speed through the eddy pressure fluctuations (Kimball and Lemon 1970-
1971, Baldocchi and Meyers 1991, Takle et al. 2004, Maier et al. 2010, Mohr et al. 2016, Poulsen et al. 2017, Mohr et al.

2017). The wind eddy pressure fluctuations generated by the above canopy penetrate below the canopy forcing the soil gases
efflux. This is a so-called pressure pumping that may be responsible of up to 50 to 100-fold enhanced effluxes. In other words,
the most relevant way to correct the chamber-based measurements below the canopy is to correct it with the pressure
fluctuations power spectrum or a Pressure Pumping Coefficient (PPC) defined by Mohr et al. 2017 measured on the soil level.
However, for the bare soil as PPC and the corelated wind speed, without the vegetation presence, wind speed measurements

remain valid for the chamber-based measurements correction.
An important experimental campaign concerning $CO_2$ effluxes measured by NSS technique is currently in progress and the
first results are showing that described methodology (several consecutives' measurements with a different fan speed in order
to deduce real efflux) is not directly applicable to other gases effluxes measurements by NSS technique such as $CO_2$ or $N_2O$,
probably because, contrary to the evaporation, the wind is mainly acting on the stock and not on the production of these gases.

However, a similar technique is not excluded.



During the calibration measurements, some interesting observations concerning sandy soil evaporation process and chiefly clayey soil evaporation process are reported.

All the soil samples under all the studied winds display two evaporation stages, the first one almost constant with a lowering

soil moisture is greatly affected by the wind, and the second one less affected by the wind with an exponential behaviour versus the soil moisture. Both soil samples display a decade of $RE$ difference without the wind and with a very small wind due to the air boundary layer perturbation.

The sandy soil does not display an observable ability to absorb the water vapor and its stocking capacities are limited. However, the apparent evaporation inertia is well conspicuous on the relatively wet soil under a relatively strong wind.

The clayey soil, on the contrary, displays a great sorption ability and water vapor stocking or unstocking capacities with a characteristic time in the hour range. The inertia is partially screened by the sorption magnitude and a strong external wind is necessary. The evaporation curve versus soil water content shows a common point $CP$ ($w = 15.5\%$ and $RE = 0.024\mathrm{gm}^{-2}\mathrm{s}^{-1}$) for every wind and soil textures but nil wind speed. The curve evaporation versus water content below the $CP$ change its slope. This point seems to correspond to the matrix air entry value MAEV.

**Competing interests.** The author declare that they have no conflict of interest.

**Acknowledgements.** This study was mainly funded by ICOS France and Anna Zawilski (private donations). I would like to acknowledge Tiphaine Tallec (CESBIO, Toulouse, France) for her useful discussions. A special thanks to Valérie Le Dantec (CESBIO, Toulouse, France) who greatly contributed to motivate this work. I am particularly grateful to Katia Bonne (LI-
COR Biosciences GmbH, Bad Homburg, Germany) and Jason Hupp (LI-COR Biosciences, Nebraska, USA) for discussions and communications.



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
