# Peer review of "Wind Influences Corrected Auto-calibrated Soil Evapo-Respiration Chamber (ASERC), Evaporation Measures."

_Geoscientific Instrumentation, Methods and Data Systems, 2021_

## Author Response (AR1)

Answer to RC1:

I am sincerely grateful to my first referee (RC1) for spending his time to carefully read and annotate my paper giving me an opportunity to improve it. I am, of course, ready to rearrange the text for more readability however, before doing it, I would like to have the second referee comments in order to do it at once.

I am answering here point by point to the RC1 questions.

RC1: "The question of the validity of the results for other atmospheric conditions (turbulence, temperature, …) could be raised and have to be argued before to accept the MS"

Effectively this question is not only legitimate but also essential. I was probably not clear enough then I am adding in the text the corresponding statements. Indeed, as the chamber technique is invasive, the conditions of the measurements may be different of the real, means "not chamber head deployed" conditions. This difference may affect the measurements. Then, for precise measurements we have to minimize the differences if we can (temperature, pressure) or to correct the measurements (wind speed). As special attention was given to the design of the chamber in order to do not affect internal chamber head temperature by solar radiation screening and IR radiation screening and do not affect the pressure variations incorporating an "expansion volume" the difference of the temperature or the pressure inside the chamber head or outside are quite similar. In other words, chamber measurements do not have to be corrected versus temperature or pressure. The initial air humidity is also assumed to be the same as the chamber fan is engaged just before the chamber head deployment in order to flush the sensor. Only the air movements cannot be preserved and its influence on the soil evaporation has to be corrected. Additionally, the calibration experience was conducted during over two years in a deserted room with great temperature variation range (temperature not controlled most of the time) and under natural pressure and air humidity variations.

RC1:" The wind velocity influence on soil evaporation, which is studied here, depends on numerous factors like the level of turbulence (friction velocity), the air temperature or the vertical component of the wind speed. Logically the difference between the real evaporation and the measurements one, obtained with a chamber having specific values for these factors, will be dependent on theses values. The results obtained would therefore not be generalizable to all weather conditions but only applicable when these factors have same value as in the experiment presented. It is therefore difficult to understand how to exploit in a general way the notion of susceptibility proposed by the author »

Concerning the Air temperature, please see my previous answer. Concerning the vertical air speed, in absence of a mass flow from the soil, the vertical mean air speed is considered as nil. In the usual conditions, the only notable mass flow results from the evaporation itself and its influence on the evaporation is negligible. The wind characteristics (friction velocity) have, indeed, a very important influence on the evaporation. However, once again, we should consider the way how the friction velocity may influence evaporation and if it can be disturbed by the chamber head deployment. I

will develop this important point in the paper. As show numerus studies of the soil evaporation (references already given in the paper), as well as described studies, on can observe three evaporation stages. In the first stage (high water content), the soil evaporation is limited by water vapor transport in the air and consequently by water vapor removal from the boundary air layer in the soil surface proximity. In this stage, the wind speed and the friction velocity have great influence.

In the second stage, where the soil evaporation is limited by the water availability, the wind speed influence is much lower but not nil.

In the third stage, the soil evaporation is extremely low and the wind, turbulent or laminar, has a very low influence on it.

In the first and second stage, the friction velocity or the turbulences contributing greatly to evacuate the water vapor from the boundary layer and from the superiors' layers maintaining air humidity low (Monin-Obukhov similarity theory). Then, as the water demand is great, the water vapor production is consequent. If the air humidity is high, with or without important friction velocity the soil evaporation is low. The turbulence (quantified by friction velocity) has great influence on the air humidity (quantified by water vapor demand) which in tourn has great influence on the soil evaporation on first evaporation stage. On the second soil evaporation stage, the water vapor demand is also important giving a maximal evaporation but does not control the evaporation rate. The boundary air layer turbulence is mainly function of soil surface shape and the wind speed; this influence has to be corrected since the wind is replaced by the internal fan blowing during the chamber head deployment. As the chamber operations protocol is optimized for initial air humidity preservation then preserving the initial water vapor demand, on can reasonably assume that the chamber measurements do not have to be corrected versus air water vapor transport ability (friction velocity).

Another consequence of the turbulence's presence are the pressure oscillations. Recent studies (references given in the paper) show that the pressure oscillations may induce important effluxes from the soil. Again, as the chamber head includes an expansion volume equalizing pressures between the deployed chamber head volume and exterior pressure, there is no correction to be made for pressure fluctuations.

Additionally, different external blower simulating the wind and having different positions was experimented without any notable difference.

We have to correct the measurements against the effects on evaporation modified by the chamber head deployment which are mainly the boundary air movements.

RC1:" Wind is used to signify wind velocity modulus, that should be corrected"

Absolutely, previously noted "wind" will be changed by "wind speed" when applicable.

RC1: "The text is difficult to read and would need an editor to improve the English writing."

An improvement of the English writing should be done.

RC1: "The manuscript should be shortened to gain in readability. I suggest removing all the text and graphs concerning the impact of the external gas analyzer and its corresponding tubing system (e.g. in section 3.1.1 about the delay + L255-269 & Fig 3 + the discussion about the start & end points in Section 3.1.2). The same for the parts not giving element for correcting or calibrating the chamber measurements (e.g. section"

This remark was obviously truncated but I do understand the point. My concern is to present an apparatus with a corresponding protocol giving to interested readers ability to reproduce it and to use it. Not only a complete description of the adopted setup is necessary but also a description what is not suitable and why. For example, most used chambers are made for external analyzer use. These analyzers, most often, providing also the water vapor concentration measurement and a natural but hazardous decision would be to use these chambers for soil evaporation measurement. The start and stop fitting point are important as it may affect significantly the results. I would like to share my experience with others in order to save their time. All this give a long paper, unusually long, I am aware. I would like to avoid a situation like for the CO2 (or GHG) measuring closed chambers, existing for over a century but with very disparate information about, leading often to unadopted use and interpretation which is the subject of my current project. The CO2 and in general GHG gases measuring chambers are also affected by the wind speed but very little literature thereon is available. To my knowledge, there is no calibrated GHG chambers yet.

RC1: "A Theory section should be created and contained all the theoretical considerations proposed in the Results & Discussion section (see Specific Comments) »

As stated at the beginning of my answer, a corresponding change will be done after the second reviewer report.

RC1: "The author presents a general comparison between real evaporation and measured ones and after explains the difficulties due to the influence of soil water content. It's not clear if the solutions proposed to overcome these difficulties (Eq 11 &13) are included in the first comparison »

I am sorry but I do not understand this point. I am discussing differences between RE and ME and difficulties due to the WIND SPEED influence. Formula 11 is giving the second stage real evaporation versus gravimetric soil water content and formula 13 describing the internal chamber head air mixing process. As mentioned on the figure legend, the comparison depicted by figure 8 concerns the real evaporation RE and the corrected measurements m*M not ME. M is a function of ME and m is a function of wind speed.

Specific Comments:

RC1: "M&M: Recall the formula for calculating the Measured flux of Evaporation ($ME$) from the temporal evolution of air moisture
L105-107: Remove the description of all the material that is not used in the manuscript"

It will be done.

RC1: "L112-114: How the additional variable volume is taken into account for the calculation of the flux"

As usually for closed chamber technique, the additional volume generated by internal volume expansion due to evaporation was not corrected as it is hardly measurable and negligible versus total chamber head volume especially for short duration deployments. The chamber head air temperature increases and corresponding air expansion should be avoided and it is one of the raised points.

RC1: "L126: There are two points of contact between the chamber support system and the
collar. Doesn't this influence the weight measurement made by the scale?"

The collar is fixed permanently on the chamber body. The bucket with the soil sample is not touching the collar and is placed on the scale with hermetic but flexible plastic foil between the bucket bottom and the scale. In this way, the internal chamber head air is entrapped in and the flexibility of the plastic foil ensures a good accuracy of the bucket's mass measurement by the scale.

RC1: "L132-133: How accurate is the measurement with the scale? Is it sufficient to have a good temporal follow-up of the real evaporation?"

The scale resolution is of 0.01g allowing then a very accurate follow-up.

RC1! "L137: The logarithmic profile is not applicable near a surface"

Absolutely, in the case of turbulent flow, the difference between the near surface and 2m high wind speed is even bigger than with logarithmic profile. The logarithmic formula was used only as an illustration for accessing wind speed order of magnitude changes with height which is absolutely not crucial for this paper.

RC1: "L141: Replace "the wind speed" by "the external wind speed""

Will be done.

RC1: "L148: Replace "fan influence" by "external wind influence""

Actually, the adapted expression would be "internal fan influence"

RC1: "L158: Add "mass" between "bucket" and "with dry""

Will be done.

RC1: "L169: This exponential rise is indeed often used but it must be specified that it is to
reflect the accumulation of a scalar of interest whose emission flux accumulates in a closed volume and with flux value depends itself on the concentration in this volume"

In this paragraph I am explaining why the exponential rise behavior is so often present in the nature by a mathematical reasoning. There is no allusion to the accumulation technique yet. Mathematically speaking, even in the case of non-accumulation the exponential rise formula is still valid $c(t)=A*exp(-k*t)+B$ with A=(equilibrium state)-(initial state)=0, then $c(t)=B$, a constant then (no accumulation).

RC1: "Eq 2-5: Why use C and then q for the concentration of the gas of interest here?"

The equation 2 is a general equation and given as explanation of exponential rise evolution and formula 5 is a particular formula used for ME calculation resulting from q (water vapor concentration) temporal variation.

RC1: "Eq 5: The equation is not correct; the measured flux depends on the volume of the chamber and the intercepted surface. The units of the left hand side are grams of water vapor per m² per second while the units of the right hand side are grams per m³ per second."

Absolutely, I have omitted a constant factor on the right side of the equation (constant versus time and equal to the chamber head volume divided by the delimited soil surface area). Will be corrected, of course.

RC1: "L230: It's not completely evident how a "simple simulation" can be realized to simulate the impact of the sensor response time, please give more information"

I can provide more details and it is very simple. How to simulate "slow" sensor and how to explain that it may provide results that lead to apparent bigger flux that it is in reality. However, it will make the text longer. Supposing the measured variable following an exponential rise law and the sensor answer to a step like measured variable evolution also following exponential rise evolution, as usually, but with not the same characteristic time, of course. By discretization of the time (means using steps Dt=constant), we can calculate for each step the real value of the variable and the corresponding measured value considering the initial state of the sensor calculated for preceding step and the new equilibrium state which is, nota benne, never reached. This simulation will provide results depending of the adopted time step Dt importance. However, making Dt progressively smallest, the calculated results beginning stable and do not change notably. We can then consider the stable results as accurate.

RC1: "Section 3.1.3 contains only theoretical considerations and should be move in a Theory section that should be implemented".

The text will be rearranged after the second referee answer.

RC1: "Fig 4: Indicate what the red and blue dots correspond to in the legend and in the corresponding text."

Will be done.

RC1: "L338: "As we can see" indicate which figure you are referring to. What is the quoted
configuration and how to see the similarity with wind influence?"

The figure used as illustration is the figure 5 (its number is given in the third sentence and will be moved to the firs sentence). The adapted configuration is the internal fan aspirating the air from the soil surface and blowing it on the chamber cloche. On the figure 5, we can see that bigger is the PWM, it means bigger is the internal fan speed and consequently bigger is the blown air speed, bigger is the corresponding ME. The similarity with the wind speed is that bigger is the wind speed and bigger is the RE. More precise description of this similarity will be added to the text.

RC1: "It's the logarithm of $ME$ and not $ME$ that is represented"

It is well ME which is represented on a logarithmic scale. The results are similar with logarithm of ME represented on a linear scale.

RC1: "Eq 8: It is difficult to understand why the average of $ME_{10}$ and $ME_{30}$ must be multiplied
by $Z$ if it is indeed the one that appears in the exponential of Eq 7"

The approach is similar to that of Rayleigh approach. As the measured fluxes ME have to be corrected because the conditions of the measurement are different from the real conditions by the wind speed suppression and internal fan blowing introduction, we have to find the correspondence between these two effects. Internal fan influence is well represented by the exponential law versus 1/PWM (empirical constatation). Bigger is the

"susceptibility" of the soil evaporation to the internal fan blowing and bigger is Z. As a working hypothesis, I have supposed that the real evaporation will be function of ME and Z then a measured evaporation ME multiplied by Z elevated to some power with a wind speed dependent factor. After numerus tests, as stated in the text, the best results were given with a mean measured evaporation (it is not the same as a measured evaporation) multiplied by Z. "Best" means that there is a good proportionality between RE and the formed function and that the proportionality factor is soil nature or texture independent. This function is purely empiric and I do not have any theoretical explanation for.

RC1: "L368: How can we deduce from the fig 7 that $m$ is not temperature dependent?"

We cannot see on the figure 7 that m is temperature independent but we can see that m is wind speed dependent and that this is the same dependance for both, sandy soil and for clayey soil which have a very different wind speed evaporation influence behavior. The clayey soil measurements and the sandy soil measurements were absolutely not performed at the same temperature and each soil sample measurements were performed during few weeks, day and night with corresponding temperature changes. We can see on the figure 8 that the correction gives good results, whatever is the temperature, pressure or initial RH.

RC1: "L394-424 (Section 3.2. and Fig 9): Why do you use $RE$ in the equation and $ME$ in the
Figure? Why do you present $ME$ values in the Figure when you have shown just before that $ME$ has to be corrected by Eq. 8 and 9 before to be considered equivalent to real evaporation? What is the benefit of this part of the manuscript in the process of correction of chamber measurements to give the real evaporation?"

The title of the section 3.2 is, indeed, misleading, indeed it should be actually "Laboratory measured soil evaporation RE results. And will be moved just before the sections 3.2.2

This section is not essential for calibration but just depict the exponential variation of RE in the second stage for both, sandy soil and clayey soil. To my knowledge, this result was not reported yet and may be interesting for soil evaporation study. The shortly presented evaporation results are describing "what we are working on" as the chamber is exactly made to measure it.

RC1: "L425-470: All the theoretical considerations and equations of the Section 3.2.1 should be move in a Theory section. It's not clear how the process describes here intervenes in the calibration of the measurements made with the chamber or in their correction"

The interest of these observations is in the understanding why ME are different from RE which is the main subject of this paper.

RC1: "Fig 11: Why show the evolution of $ME$ after the wind has stopped rather than showing values of $ME$ for different wind values if it is to discuss the influence of the latter on $ME$?"

I have already discussed on the influences and in that section, I am discussing on why the ME behavior is very different for sandy soil and for clayey soil. When the chamber head is deployed, the external wind influence is stopped and the internal fan influence is started. Sandy soil adapts almost immediately to the new conditions but not clayey soil. This point is important for later measurements of clayey soil evaporation under changing wind speed as the measured results correspond to the pondered integration of previous wind speed influences not only to the actual wind speed influence. This fact will be added more explicitly to the text.

RC1: "L 525-528: Are the different values of the proposed constants valid for all PMW?"

Absolutely not. The effective mixing time cannot be the same for all PWM ratio.

RC1: "L565-610 (section 3.2.2): This part is not necessary since it refers to the influence of cracks on real evaporation but doesn't bring new element for correcting or calibrating the chamber measurements."

Indeed, this section is not necessary for the calibration process and it is reported as finding during calibration studies. Again, to my knowledge, the common point was not reported elsewhere and it may be interesting for the soil studies. Only two points concerning RE are shortly reported at the margin: exponential behavior of the evaporation for both soils on the second evaporation stage and a common point existence in the clayey soil. These points coming from the described calibration studies as for chamber calibration we need the chamber evaporation measurements (ME) and the real evaporation measurement (RE). I think that there is no reason to hide the real evaporation measurements even if it presents some particularity not yet signaled.

Complements of answer to RC1:

With remarques of the RC2 referee, I was able to reformulate this paper.

A more detailed description of required corrections is provided. A figure depicting RE versus not corrected ME is added.

M&M include now the used algorithms, sensors use and every information related to the materials and used methods. The laboratory measured RE was moved to an appendix. The description of a slow sensor simulation and of the resulting possible evaporation overestimation is also moved to an appendix but shortly developed (one figure dedicated for).

Answer to RC:

I would like to thank my second referee (RC2) for his work to truly read my text and for his suggestions which are all embedded in the revised version.

I would like to answer point by point for particular suggestions or questions. Concerning all the typesetting's suggestions and corrections it will be done in a revised version so, I am not answering it individually.

CR2: "I found many difficulties in reading the paper, both for the English style (which, to my opinion, needs extensive revision) and for the overall structure of the work."

I do agree to revise the structure and to correct English writing by a professional. In the new structure, in response to both, CRC1 and CR2, Laboratory findings on RE are moved to an appendix. A more detailed description of slow sensor simulation and possible resulting overestimation is moved to another appendix in order to shorten the "conclusions" section. Also, description of used algorithms and description of a correct sensors use is moved to M&M section. All the changes make "Conclusions" shorter.

CR2: "Abstract should be focused more on the goals and the structure of this research, providing then specific highlights on the main findings and conclusions. Introduction should provide a general background on the current state of the art, leave at the disposal

of the reader all the elements for contextualize the topic and what will discuss later. As an example, Par. 1 could be moved to this section."

Effectively the structure was rearranged.

CR2: "Par. 2.4: is it necessary to add a paragraph for a few sentences?"

It was merged with previous section and moved to M&M section.

CR2: "Par. 3: the nested structure of this paragraph heavily affects the readability of the paper. Furthermore, I would appreciate if the whole paragraph should be reformulated in a shorter form."

This section is much shorter in the new version.

CR2: "Please, check the compliance of formal use of unit of measurement to the International System"

In general, I am always following the USI except when measured scalars are better described with others units. Example: water fluxes USI would be in $kg/m^2/s$ but it will be too small so I am adopting rather $g/m^2/s$. PWM is unitless but expressed in % as every PWM generator require inputs as %. Consequently, $Z$ is also unitless and expressed in %. W, gravimetric soil water content, is unitless too and also expressed in % as we always converting it to % rather than conserving it in Kg/Kg or g/g (number comprised between 0 and 1).

CR2: "A table of acronyms and unit of measures can be a precious support for the readers"

Effectively, as I am using lot of acronyms and a list is welcome. I am adding it.

R2: "Line 22: are you sure that "WMO168 2008" is the right way to cite the reference?"

Reference changed.

CR2: "Line 40: could you add a general framework related to eddy covariance technique?"

Done

CR2: "Line 51: Zawilski in progress…what is this reference?"

Not yet and I have withdrawn it.

CR2: "Line 61: why reporting this information about the maize? Is it necessary for the discussion?"

I am citing it as illustration to show that evapotranspiration measurement and then some "models" use in order to split it to soil evaporation and vegetation transpiration gives results that are model dependent. And models are numerus.

CR2: "Lines 121-122: again, are you sure that is the right way to cite the reference?"

These lines describing the used programming language (Labview) and its designer owner.

CR2: "Lines 170-183: is it necessary to report the mathematical description of the solution?"

Basing on my personal experience and exchanges with my colleagues, I found useful to recall why a physical phenomenon with variations proportional to difference between actual state and equilibrium state displays an exponential rise behavior. And this explanation resides in a mathematic consideration. The exponential rise behavior is widely present in the nature and appears several times in this paper (water vapor accumulation, slow sensor measurements, soil water vapor sorption….)

Line 187: please, check Eq. 5

Absolutely, as answered to my CRC1, I have corrected it by adding factor V/S (chamber volume by collar surface) on the right side and completed by the initial slope calculation.

CR2: "Line 211: could you provide references to this sentence?"

Not yet as it is always my current work and consequently, I have changed "are" to "will".

CR2: "Line 241: tau_63 stands for…?"

As stated in the concerned sentence, it is à "response time". This notation, is very often used for sensors characteristics description. For better description it is listed in the "often used acronyms and unit" list.

Line 242: RH is Relative Humidity?

As described in the text, yes. Listed in the acronyms and unit table.

Complements of answer to CRC1:

With remarques of the CR2 referee, I was able to reformulate this paper.

A more detailed description of required corrections is provided. A figure depicting RE versus not corrected ME is added.

M&M include now used algorithm, sensors use and every information related to the materials and used methods. Laboratory measured RE was moved to an appendix. Description of slow sensor simulation and of the resulting possible evaporation overestimation is also moved to an appendix but shortly developed (one figure dedicated for).